# Insights into Emergence of Antibiotic Resistance in Acid-Adapted Enterohaemorrhagic *Escherichia coli*

**DOI:** 10.3390/antibiotics10050522

**Published:** 2021-05-02

**Authors:** Salma Waheed Sheikh, Ahmad Ali, Asma Ahsan, Sidra Shakoor, Fei Shang, Ting Xue

**Affiliations:** 1School of Life Sciences, Anhui Agricultural University, Hefei 230036, China; salmawaheed@ahau.edu.cn; 2School of Agronomy, Anhui Agricultural University, Hefei 230036, China; ahmadali@ahau.edu.cn; 3Faculty of Life Sciences, University of Central Punjab, Lahore 54000, Punjab, Pakistan; asmaahsan23@gmail.com; 4Station de Neucfchateau, CIRAD, 97130 Sainte-Marie, Capesterre Belle Eau, Guadeloupe, France; saammii03@yahoo.com

**Keywords:** foodborne infections, enterohaemorrhagic *Escherichia coli*, acid tolerance, cross-protection, multidrug resistance, two-component signaling system

## Abstract

The emergence of multidrug-resistant pathogens presents a global challenge for treating and preventing disease spread through zoonotic transmission. The water and foodborne Enterohaemorrhagic *Escherichia coli* (EHEC) are capable of causing intestinal and systemic diseases. The root cause of the emergence of these strains is their metabolic adaptation to environmental stressors, especially acidic pH. Acid treatment is desired to kill pathogens, but the protective mechanisms employed by EHECs cross-protect against antimicrobial peptides and thus facilitate opportunities for survival and pathogenesis. In this review, we have discussed the correlation between acid tolerance and antibiotic resistance, highlighting the identification of novel targets for potential production of antimicrobial therapeutics. We have also summarized the molecular mechanisms used by acid-adapted EHECs, such as the two-component response systems mediating structural modifications, competitive inhibition, and efflux activation that facilitate cross-protection against antimicrobial compounds. Moving beyond the descriptive studies, this review highlights low pH stress as an emerging player in the development of cross-protection against antimicrobial agents. We have also described potential gene targets for innovative therapeutic approaches to overcome the risk of multidrug-resistant diseases in healthcare and industry.

## 1. Introduction

Low, acid pH kills bacteria. Humans have evolved acid conditions to protect from food-borne pathogens, and pathogens have evolved ways to bypass those protections. Acid tolerance, the ability to survive acid conditions, is a clinically important phenotype of foodborne pathogens and an overwhelming issue for public health [1,2]. Most microorganisms, including pathogenic bacteria, prefer to grow at pH 6–7 [3]. Globally, the use of acid-based antimicrobial agents is widely practiced because it kills microbes at workplaces, in hospitals, on medical equipment, as food preservatives, in soil niches and during wastewater treatment [2,4]. Commercially, wastewater is treated with nitrous acid (disinfectant) while diluted acetic acid, hypochlorous acid, chlorhexidine, ethanol, acetate, hydrogen peroxide and boric acid are commonly used to treat wounds and infections in hospitals [5,6,7]. Acidified-chlorinated water (a blend of hydrochloric acid and mild organic acids) is often sprayed on meat and lettuce leaves to attain a pH of 2.5 [8,9]. This helps kill certain Enterohaemorrhagic *Escherichia coli* (EHEC), which causes severe poultry, bovine and human extra-intestinal diseases at a very low infectious dose (10–500 cells) [10,11].

EHECs colonize farm animals and are easily transmitted to humans, especially through poorly cooked beef and raw milk. The uncontrolled use of inappropriate veterinary antibiotics and growth promoters during animal husbandry and farming is a major reason for the spread of EHEC strains [12]. Besides, excessive administration of disinfectants, inadequate hygienic practices and contaminated meat and/or dairy products also facilitate the emergence of acid-tolerant strains. These EHECs can survive in diverse acidic environments (pH 2–3) including soil, farm water, apple cider, meat and even in the human gastrointestinal tract [2,13]. The ability to survive extremely acidic gastric fluid increases the risk of foodborne diseases in humans caused by EHECs [14,15]. This adaptation not only provides survival opportunities but also facilitates several cross-protective benefits, including enhanced antimicrobial resistance, biofilm formation, pathogenic adhesion, and colonization [16,17].

More than 400 serotypes of EHECs are known to cause several life-threatening diseases, such as hemorrhagic colitis, intussusception, bloody diarrhea, inflammatory bowel disease, and systemic hemolytic uremic syndrome (HUS). HUS is a multi-symptomatic syndrome caused by the highly prevalent serotype O157:H7. Infected patients suffer from thrombocytopenia, acute renal failure, and hemolytic anemia leading to death [18]. In severe circumstances, multi-organ failure has been reported [18]. EHECs epitomize overwhelming health concerns, especially in newborns, children, and immunocompromised patients; these patients suffer high rates of morbidity and mortality [19]. Globally, the well-reported virulent serotypes of EHECs are O26:H11, O45:H2, O103:H2, O111:H8, O121:H19, O145:H28, O157:H7, and O157:H. Among these, EHEC O157:H7 strains are highly virulent and have been found responsible for several foodborne outbreaks across the globe having a mortality rate of 5% [20,21,22,23]. More importantly, newly emerging EHECs, especially EHEC O157:H7 and EHEC O80:H2 serotype strains, are multidrug-resistant (MDR-EHECs); no effective antibiotic has been reported for treating diseases caused by these strains [18,21,24,25]. EDL933, a well-known EHEC O157:H7 strain, first reported in 1983, affected 47 people in Oregon and Michigan [26]. So far, this deadly strain is reported to cause 73,000 illnesses, 2200 hospitalizations, and 60 deaths annually in the United States [27]. In addition, the EHEC O80:H2 strain was reported in France where it caused severe symptoms of HUS in association with bacteremia; all antibiotics were ineffective [25]. Similarly, another EHEC serotype, O104:H4, was also found resistant to almost all known antibiotics in Europe; it resulted in 3800 disease cases and 53 deaths [25,28]. These strains possess multidrug-resistance-encoding regions that provide enhanced resistance to several known antibiotic determinants, including chloramphenicol, aminopenicillin, cefotaxime, neomycin, aminoglycoside, sulfamethoxazole, nalidixic acid, beta-lactams, cotrimoxazole, amoxicillin-clavulanic acid, imipenem, norfloxacin, tetracycline, phenicols, streptomycin, trimethoprim, ciprofloxacin, kanamycin and carbapenems [18,19,29,30,31,32,33,34].

Antibiotic resistance genes can be transmitted from animals to humans in several ways; for EHECs, the foodborne route is probably the most important. In EHECs, the multidrug-resistance genes are encoded by plasmids that can easily be transmitted from one organism to another through horizontal gene transfer or bacteriophages, resulting in the emergence of zoonotic infections in humans [18,19,30]. Using antibiotics for the treatment of EHEC infections has long been controversial as several studies correlate the use of antibiotics with death and an increased rate of cerebellar hemorrhage [25,35]. It has been suggested that antibiotics targeting DNA synthesis (quinolones and ciprofloxacin) should not be used during acute EHEC infections, as treatment increases several systemic complications [36,37]. For example, treatment with fluoroquinolones induces Shiga toxin secretions, resulting in even higher mortality rates [25,36,38,39]. These complications are highly dependent on several factors including, type of antibiotic used, dose of antibiotic, time of administration, route of antibiotic administration, type of EHEC strain and severity of the infection [35].

EHECs are widely distributed in domestic ruminants (sheep, goats, pigs, turkeys and cattle) and use food as a vector to infect humans. Designing effective intervention technologies and risk-management options are required to overcome antimicrobial resistance in the food chain. These foodborne pathogens suffer from various physical and chemical stresses during cooking, such as heating, freezing, acid, and salt treatments [40]. These treatments can efficiently kill certain pathogens; however, pathogens that survive these treatments become genetically and physiologically strongly adapted. Fecal contamination of water and ingestion of EHEC-contaminated food products (meat, milk, raw vegetables) create a greater risk of transmission of resistant genes from pathogenic bacteria to commensal gut flora [41,42]. However, the extremely acidic pH of the mammalian gastrointestinal tract can kill almost all types of microbes, except acid-tolerant microbes. They can then transfer resistance genes to microbial flora in the gut [43]. EHECs are well known for their adaptation to the acidic environment; their tolerance level is comparable to that of acidophiles. Furthermore, foodborne, acid-adapted strains also confer cross-protection to antibiotics, which plays a vital role in the spread of multidrug-resistant pathotypes [16,44,45,46,47,48]. The human gastrointestinal tract is reported to provide the best environment for the emergence, transmission, and spread of antibiotic-resistance genes in bacterial populations. Several factors assist this transmission of genes from one bacterium to another including high cell density, antibiotic exposure and innate ability of gene transfer [43]. The human body appears to serve as an “antibiotic resistance gene bank” to generate resistant pathotypes, which may emerge as a great public health challenge [49,50].

All EHECs are highly acid-tolerant, especially, EHEC O157:H7 strains, which are known to be the best-adapted strains. The resistance potential of all EHEC O157:H7 strains against almost all marketed antibiotics is posing a challenge to treat life-threatening diseases caused by these strains. These strains employ sophisticated antibiotic inactivation, structural modifications, target replacement and antibiotic efflux activation mechanisms [19]. There are several ways through which different antimicrobial agents target intracellular processes by blocking the binding proteins, affecting cell division, modifying ribosomal proteins, and causing competitive inhibition [51]. In penetrating the cell, these antimicrobial agents first need to breach the outer membrane of the bacterial cell [52].

Several antimicrobial agents tend to kill pathogens by affecting the integrity of the outer membrane. The outer membrane of Gram-negative bacteria is composed of a negatively charged hydrophobic lipid bilayer (lipopolysaccharides) and pore-forming proteins. The negatively charged lipopolysaccharides can easily be disrupted by positively charged antimicrobial peptides (cationic antimicrobial peptides). Therefore, lipopolysaccharide modification provides a way to protect against outer-membrane disruption [52].

The involvement of various two-component systems (TCS) and global regulators also facilitate the expression of various efflux pumps and competitor proteins. Microorganisms use efflux pumps to regulate their internal environment by eliminating harmful compounds such as metabolites and antimicrobial agents [53]. In acid-adapted EHECs, the activation of multiple efflux pumps is one of the major strategies for developing antimicrobial resistance [54,55,56,57]. Understanding the relationship between acid tolerance and genetic adaptability demands detailed insight into cellular responses to changing environments [54,58]. This review examines the expression of genes that are induced by acidic pH and how acid tolerance facilitates antibiotic resistance.

## 2. Acid Tolerance Potential of EHECs

Extremely virulent serotypes of EHECs (O157:H7 strains) have regulatory aspects of acid tolerance that are not found in other *E. coli* strains [59]. Although EHECs can acquire antibiotic resistance by horizontal gene transfer, they can also develop *de novo* resistance during exposure to various environmental stresses, especially low pH. The mechanisms that initially allow the bacteria to survive stress subsequently result in resistance to even higher antibiotic concentrations measured by minimum inhibitory concentration (MIC) [60]. The set of genes involved in protecting against acid stress are also responsible for the acquisition of antibiotic resistance [51]. Thus, it is important to understand the regulation of several transcriptional regulators (GadE, H-NS) and two-component signal transduction kinases (EvgAS, PhoPQ, RcsB) that are activated in response to low pH. At the molecular level, key players involved in mediating acid tolerance include enzymatic cascades of specific decarboxylases and families of two-component signal-transduction kinases.

The signal-transduction kinases consist of a sensor kinase and a response regulator that controls acid-tolerance, pathogenicity, and antibiotic resistance [61,62,63,64]. These two-component kinases work in coordination with multiple global regulators, transcription factors, and several other local regulatory proteins [1,14,65,66]. This regulation further involves several regulatory proteins, chaperons, and periplasmic proteins that protect the EHECs against DNA damage and protein coagulation.

Four main systems regulate acid tolerance: oxidative, glutamate-dependent, arginine-dependent and lysine-dependent acid resistance systems. They work in tandem to protect cells from acid stress (Figure 1; Table 1) [67,68,69,70,71,72]. These mechanisms exchange intracellular protons for an amino acid (glutamate, arginine, or lysine) and expel amines into the extracellular media in exchange for the corresponding amino acid [73].

The oxidative system does not involve an externally-derived amino acid; it is regulated by sigma factor RpoS and catabolite repressor protein (CRP). It provides the lowest level of protection at pH 2.5 at the expense of energy, as shown in Figure 1 [74]. The glutamate-dependent system is adapted to protect under extremely acidic conditions and is highly efficient. The arginine-dependent system is only induced under anaerobic conditions. It requires arginine decarboxylase (AdiA) and an arginine:agmatne antiporter (AdiC) to provide a modest level of protection under mild acidic conditions (Table 1) [75]. The lysine-dependent system also works under slightly acidic environments; the efficiency of resistance is lower than seen with other systems [68,76]. This system involves a lysine decarboxylase (CadA) and a bifunctional lysine:cadaverine antiporter (CadB) (Table 1). The pH inside certain compartments of the mammalian gastrointestinal tract drops below 2; which renders all acid resistance systems inactive except the glutamate-dependent acid one.

**Table 1 antibiotics-10-00522-t001:** Overview of the genes involved in acid resistance regulatory systems in *E. coli*.

Protection Mechanism	Main Substrate	Decarboxylases	Antiporter	Final Product	Regulators	Level of Protection	pH	Reference
Oxidative system	Glucose	-	-	-	RpoS	Least	2.5	[77,78,79]
Glutamate dependent system (GAD)	l-Glutamate	GadA*,* GadB	GadC	GABA	GadE*,* GadX*,* GadW	Highest	≤2	[67,75,79,80,81,82]
Arginine dependent system (ADI)	l-Arginine	AdiA	AdiC	Agmatine	-	Modest	5.2	[75,79]
Lysine dependent system (CAD)	Lysine	CadA	CadB	Cadaverine	CadC	Quite ineffective	NA	[75,79]

Note: All abbreviations are defined at the end of the manuscript.

### 2.1. Glutamate-Dependent Acid Resistance System

The glutamate-dependent system provides the highest level of protection under extremely acidic conditions. This system involves two glutamate decarboxylases, GadA and GadB, that work in coordination with the gamma-aminobutyric acid (GABA) antiporter (GadC) and a set of these three genes known as GAD [83]. Extracellular glutamate is exchanged with intracellular GABA through the GABA antiporter GadC and subsequently decarboxylated by GAD [84]. During the decarboxylation of L-glutamate, the α-carboxylic group is released as carbon dioxide and a proton is incorporated into the GABA molecule, which is exported across the inner membrane in exchange for more glutamate through GadC [67]. In addition, the antiporter increases the availability of glutamate to the GAD enzymes, thereby, enhancing the efficiency of the system by acidifying the cytoplasm [85].

The functional side chain of glutamate imported by GadC has a p*K*_a_ of 4.1. Before entering the cytoplasm, during acid stress (pH 2.5), this side chain gets more than 50% protonated and these protons dissociate to acidify the cytoplasm. Therefore, the cytoplasmic pH drops to 3.6, which is an optimal pH for glutamate decarboxylase while rendering arginine and lysine decarboxylases inactive, as their optimal pH is 5.25 and 5.5, respectively [86,87]. This defense strategy works by reversing the membrane potential to maintain more protons inside as compared to the external environment [76]. The inner membrane potential remains more positive and gradually slows the flow of protons into the cell, thereby maintaining homeostasis.

### 2.2. Control of Glutamate-Dependent System

The gap between an environmental stimulus and gene regulation is bridged by sensors and regulators of two-component systems. A two-component system typically consists of a sensory kinase that monitors the environmental conditions and modulates phosphorylation of the respective response regulator. The response regulator then regulates gene expression, which changes the behavior of the bacterial cell. To cope with acid stress in the gastrointestinal tract, several two-component systems play specific roles in maintaining homeostasis and cell integrity. The selection of resistance mechanism depends upon the energy source and extracellular environmental conditions.

The protection conferred by the glutamate-dependent system is significantly higher than the other systems, allowing up to 80% survival. As a consequence, the glutamate-dependent system is considered a key player in acid regulation. This system comprises a complex network of two-component regulators with a wide array of interactions to cope with mild to extreme acid stress. The key interacting regulators of this network are GadE, EvgAS, PhoPQ, YdeO, GadW, RcsB, and GadX, which regulate gene expression spatially and temporally (Figure 2) [44,67,87,88,89,90,91,92,93,94,95,96,97]. This system is activated either by mild acidic pH during the exponential growth phase or by entry into stationary phase. Two-component systems are regulated by the induction of mild acid shock, while the *rpoS-gadX-gadY-gadW* circuit is activated during stationary phase. Once triggered, this system activates a cascade of regulatory genes that then activate the central regulators GadE and YdeO (Figure 2) [87,95,98,99]. The increased expression of central regulators results in the activation of several acid-resistance genes at different loci. This activation involves more than 20 proteins (CRP, Dps, EvgA/S, GadE, GadX, GadW, H-NS, Lon, PhoP/Q, RNaseE, sigma factor 70, sigma factor RpoS, SspA, TrmE, TopA, TorS/R and YdeO) and several non-coding RNAs (DsrA, GadY, and GcvB) (Figure 2) [91,99,100,101,102,103,104,105,106,107,108].

The sensor kinase EvgS detects the low pH signal and activates response regulator EvgA, which then starts a gene transcription cascade leading to activation of the *ydeP–safA–ydeO* circuit. The activated YdeO increases the expression of GadE and other genes involved in the regulation of acid-fitness-island (AFI) genes, namely, *slp–dctR–yhiD–hdeB–hdeA–hdeD–yhiU–yhiV–gadW–gadY–gadX–gadA.* It also activates glutamate-dependent acid-resistance genes, namely, *gadA, gadB*, and *gadC* (Figure 2). Activation of these genes and *gadE* requires the heterodimerization of RcsB with GadE. Activation of GadX also stimulates the LEE (locus of enterocyte effacement) to mediate acid-induced regulation of pathogenic traits including biofilm formation, multidrug resistance, and enhanced colonization, as shown in Figure 2.

#### 2.2.1. EvgAS: An Acid-Resistance Regulator

EvgAS is indispensable for protecting against low pH through a range of interacting mechanisms that depend upon the conditions (stress in exponential phase or entry into stationary phase) [90]. Under acid stress, the sensory kinase EvgS phosphorylates the response regulator EvgA. The activated EvgA then phosphorylates a transcriptional regulator YdeO (the AraC/XylS super-family transcriptional regulator). This phosphorylation depends on a small membrane protein SafA (sensor-associating factor A) from the *ydeO-safA* operon. The phosphorylated YdeO ultimately activates GadE, which regulates various decarboxylases and provides resistance against acid stress. It also regulates several other cellular processes as shown in Figure 2 [109]. EvgA strongly binds to the promoter regions of several genes involved in the regulation of acid resistance, such as *ydeP*, *safA*, *yfdX*, *frc*, *yegR*, and *gadE*. The role of YdeP, YfdX and YegR needs to be investigated in the context of acid resistance.

#### 2.2.2. PhoPQ: Role in Acid Regulation

PhoPQ is a two-component signaling system that responds to multiple environmental stimuli, including low pH, osmotic shock, low concentration of divalent cations and antimicrobial peptides (AMPs) [103,110,111]. It consists of a histidine kinase PhoQ that interacts with SafA and starts a phosphorylation cascade resulting in activation of the response-regulator PhoP. During the exponential phase, transcription factor PhoP activates IraM, which then interacts with RpoS [112]. RpoS is a central regulator of the stress that encodes sigma factor-38 and allows the cell to survive environmental challenges [113]. Due to this interaction, the level of RpoS increases and subsequently recruits RNA polymerase (RNAP) to RpoS-regulated promoters, including the *gadE* promoter [114,115]. Mg^2+^ stress concurrently activates PhoPQ and other regulatory proteins, thereby enhancing pathogenesis by increasing pathogen survival [94].

#### 2.2.3. RcsB: An Essential Activator/Repressor

RcsB is a response regulator that functions both as an activator and a repressor. It works in coordination with GadE to form a heterodimer on the GAD box that activates transcription of acid-resistance genes (Figure 2). It is an essential element for *gadA* and *gadB* promoter activity [82,83,84,85]. All acid-resistance promoters activated by GadE are also dependent on RcsB for their activation; the regulation mechanism of RcsB is still unknown.

## 3. Cross-Protection in EHECs

Cross-protection is the defensive adaptability of a strain when exposed to certain environmental stresses, including acid stress. Cross-protection mechanisms are either non-specific for the choice of substrate (multidrug efflux pumps), share a few common regulatory sets of genes (glutamate-dependent pathway genes), or undergo structural modifications (lipopolysaccharide chain modification) [116,117]. Foodborne EHECs encounter several acidic treatments from farm to gut and gradually adapt. This exposure to acidic conditions helps them develop cross-protection against other environmental stresses, including antimicrobial agents [118]. In addition to EHECs, acid-adapted pathogens other than EHEC pathotypes’ have also been reported in several major outbreaks all over the world, as shown in Table 2. This cross-protection poses serious concerns to the consumers (humans) and could lead to the emergence of new pathotypes.

## 4. Metabolic Adaptations

Once ingested, EHECs experience severe environmental challenges including extreme pH fluctuation and nitrosative stress (nitric acid) from volatile organic acids formed as a result of anaerobic fermentation in the gastrointestinal tract [30]. Mostly, EHECs favor pH 6–8 for their growth; to survive low pH stress they develop a transmembrane gradient to maintain homeostasis [2,126,127]. When grown at acidic pH, genes involved in metabolism, energy production and class I heat shock proteins are down-regulated to lower metabolic cost [128]. These strains consume intracellular protons through amino acid decarboxylation during acid stress, which highly acidifies the cytoplasm, resulting in increased acid tolerance [44,76]. During oxidative respiration, electron transport of membrane-bound systems, including the *atp* operon, is down-regulated to inhibit the import of protons [129]. While in the intestine, the enhanced expression of the Long Polar Fimbriae gene (*lpf-2*) mediates bacterial colonization in response to anaerobic nitrosative stress [126]. Genes involved in motility, type III secretion system (T3SS), bacterial chemotaxis, biofilm formation, adhesion, iron uptake and oxidative resistance are upregulated [1,127]. Cellular adhesion capacity (the intimin gene *eae*) of EHEC O157:H7 is enhanced by the histone-like, nucleoid-associated H-NS protein that regulates bacterial fitness and uncontrolled virulence [1,30]. In addition, the expression of *fur,* which is involved in iron uptake, is also up-regulated. In particular, low pH helps bacteria survive acid stress by enhancing motility, adhesion, and iron utilization, thereby assisting the pathogen in enhancing apoptosis of epithelial cells and become more virulent [2]. This mechanistic regulation helps the pathogen achieve homeostatic balance by modifying metabolic pathways at the cost of energy generated from redox or ATP-driven reactions.

## 5. Acid-Adaptive Antibiotic Resistance Strategies

Generally, growth-inhibiting stresses, such as low pH, high temperature, or nutritional deficiency, induce several metabolic rearrangements at the cellular and metabolic levels that influence differential regulation of more than 500 genes to ensure tight homeostasis. In response to extracellular acid stress, pathogens undergo several cellular and global transcriptional changes that alter their responsiveness to a wide array of antibiotics. These regulatory changes help the organism adapt to extreme environmental stresses and subsequently enable cross-protection consistent with the survival of the organism [44,60].

While passing through the gastrointestinal tract, EHECs experience anaerobic conditions and nitrosative stress that trigger enhanced expression of the multidrug efflux genes (*mdtEF*) and several two-component signaling systems including EvgAS, PhoPQ, RcsB, PmrAB, ArcAB, BaeSR, KdpA, and CpxAR [44,130,131,132]. These two-component systems combat cell envelope disruption caused by proton imbalance and antibiotic-induced accumulation of mistranslated peptides that can cause cell damage by disturbing homeostasis [133]. Multidrug-resistance efflux pumps are essential for withstanding antibiotic challenges and other environmental toxins. During anaerobic conditions, the global transcription factor ArcA increases the expression of MdtEF (more than 20 fold), which dramatically enhances efflux activity leading to antibiotic resistance [130]. Incubation at low pH also aids the development of antibiotic resistance, which persists even after environmental conditions shift [16,60]. Overall, these stress-induced genetic alterations confer genetic plasticity that results in enhanced population diversity, strengthening of the envelope and resistance to a wide array of antibiotics.

### 5.1. Acid-Adaptive Structural Modifications

Gram-negative bacteria have a highly asymmetric outer membrane with phosphatidylethanolamine lipids at the inner side, while the external side is enriched with lipopolysaccharides [134,135,136]. The lipopolysaccharide membrane is composed of three vital parts, including a gel-like hydrophobic anchor (lipid A), branched oligosaccharides (core region), and polymer of repeating saccharide subunits (O-antigen) (Figure 3) [135,136]. The structure of this membrane is enriched with many phosphoryl and carboxyl groups bridged with divalent cations that facilitate low permeability and antibiotic resistance [134,136].

#### 5.1.1. Acid-Induced LPS Modification

PhoPQ upregulates the transcription of acid-resistance genes under acidic stress; the same set of genes also mediate LPS modifications in EHECs [103,137]. Several environmental changes, such as acidic pH, osmotic stress, low concentration of divalent cations (Mg^2+^), and the presence of antimicrobial peptides (AMP), trigger this pathway [103,134]. EHECs have evolved this defensive strategy to remodel the outer membrane by adding a palmitoyl chain, a hydroxyl group, and a positively charged aminoarabinose sugar moiety to the lipid A anchor, acetylation of the O-antigen and hydroxylation of fatty acids through the PhoPQ two-component regulatory system (Figure 3) [111,137]. These induced modifications help EHECs become more virulent by increasing cationic antimicrobial peptide resistance and suppressing TLR4 immune responses. They also increase permeability to large lipophilic agents [138]. EHEC serotype O157:H7 is reported to develop increased resistance to cationic antimicrobial peptides, especially Polymyxin B, in response to acid stress, bile salts, and ferric ions in the human gut (Table 3) [103,137].

#### 5.1.2. Acid-Induced Antimicrobial Resistance by RcsB

The bacterial cell envelope comprises outer and inner membranes that act as a protective barrier. The outer membrane is an asymmetric bilayer of phospholipid and lipopolysaccharides (LPS). There is a thin peptidoglycan layer lying in the periplasmic space between the outer and inner membrane of the cell. EHECs are at high risk of losing cell envelope integrity and improper protein folding under extremely acidic conditions due to excessive osmotic pressure [139,140,141]. Penicillin-binding proteins (PBPs) keep adding new subunits to the outer membrane during cell growth and repair. Genetic profiling of EHECs confirmed that pH stress inhibits penicillin-binding proteins, which in turn activates the *rcs* phosphorelay to retain envelope integrity and develop resistance to amdinocillin (mecillinam) and cefsulodin (a member of the beta-lactam group of antibiotics) [141]. When acid stress is encountered, the expression of *ugd* is increased and incorporates a 4-amino acid modification to the lipid A anchor (Figure 3) [142]. The expression of PagP is also increased, which regulates lipid A palmitoylation, thereby limiting bacterial recognition by the host immune response. Activated PagP also triggers the expression of RcsB-GadE-regulated genes *cpsB*, *rprA, gadA* and *gadB* [96,142]. The Rcs phosphorelay cascade is widely distributed among EHECs; knockout mutants of *rcsB* are hyper-susceptible to beta-lactams, which suggests that RcsB is a global regulator of cell envelope integrity [139,140,141].

#### 5.1.3. CpxAR-Mediated Peptidoglycan Cross-Linking

The Cpx-TCS (conjugative pilus expression) is a well-studied TCS that counters cell envelope perturbations. Most induction stimuli for Cpx include misfolded proteins, alkaline pH, salt, changes in lipid composition and attachment to abiotic surfaces. Acid-induced activation of Cpx regulates proton influx and cell wall stability by influencing membrane porins and cross-linking between lipopolysaccharide and peptidoglycan in the outer membrane [143]. Under low pH stress, the activated CpxRA upregulates the expression of several proteins including CydAB, GadAC, CadA, and HdeABD [143]. Most of the activated genes are controlled by the glutamate-dependent acid resistance system that extends the function of Cpx acid-induced tolerance.

During acid stress, several Cpx-regulated proteases, multidrug efflux genes and peptidoglycan amidase genes also reduce susceptibility to cationic antimicrobial peptides (polymyxin B), aminoglycosides (kanamycin), novobiocin and beta-lactams [141,144,145]. Knockout mutants of the Cpx two-component system reduce susceptibility towards antimicrobial agents, as shown in Table 3 [141,145].

### 5.2. Target Replacement

#### PhoPQ- and PmrAB-Mediated Competitive Inhibition

During stress, PhoP regulates the transcription of several stress-responsive and virulence pathway genes, including *pagL*, *pagP*, and *pmrD,* to induce modifications in the lipid A anchor (Figure 3). PmrD is a small regulatory RNA that triggers PhoPQ-mediated activation of PmrAB [111,146,147]. Under acid stress, histidine and glutamate residues of PmrB sense low pH and phosphorylate PmrA. Activated PmrA triggers the *arn* operon and *eptA,* which then modify the aminoarabinose and phosphoethanolamine residues in lipid A, respectively [111,148]. In parallel, PmrR blocks the regulatory domain of lipid A phosphotransferase LpxT (a competitive inhibitor of EptA) and facilitates EptA-mediated phosphoethanolamine modification. These PmrA-dependent modifications confer resistance to cationic antimicrobial peptides, including polymyxin. WD101 (a *pmrA* mutant strain) showed 40-fold lower susceptibility to polymyxin compared with its isogenic parent, W3110 [111,149].

DNA microarray studies confirmed that coupling of the above mentioned two-component systems results in significantly reduced susceptibility for polymyxin B and colistin (both are membrane-disrupting CAPs) [51,137]. Polymyxin B and colistin are last-resort antibiotics for multidrug-resistant EHECs [51]. The lipopolysaccharide membrane serves as the first site for interacting with, as well as combating against, cationic antimicrobial peptides. Thus, the two-component systems trigger genes responsible for structural modification that prevents binding of antimicrobial peptides.

### 5.3. Acid-Adaptive Activation of Drug Efflux Pumps

In Gram-negative bacteria, multidrug-resistant efflux pumps play an indispensable role in exporting toxins or harmful metabolites and antimicrobials of different families across the inner and outer membranes. Thus, efflux pumps decrease intracellular drug concentration.

#### 5.3.1. Activation of EvgAS-Regulated Drug Efflux Genes

EvgAS regulates multiple regulatory mechanisms including acid tolerance, drug efflux transporters and bacterial drug-resistance pathways. Extracellular acid stress leads to cytoplasmic acidification that permits EvgA to activate *emrKY*, *mdtEF*, *mdfA*, *tolC* and *acrAB* drug efflux (TolC-dependent pumps) genes [150,151]. Low pH induces the expression of *emrAB* and *emrKY* multidrug-resistance efflux genes, which confer a growth advantage and facilitate multidrug resistance against extended-spectrum β-lactamases (ESBLs) (Table 3) [150,151,152,153]. In addition to the development of resistance to ESBLs, the *evgA-ydeO-gadE* regulatory cascade also facilitates antimicrobial tolerance to other drugs, including gallium nitrate (GaNt) [91]. Gallium nitrate is an FDA-approved drug widely used for the treatment of carcinogenic hypercalcemia and is effective against several clinically significant MDR bacteria. Advanced genomic techniques confirmed the gain-of-function mutation in the EvgSA two-component system in these tolerant strains [91]. Deletion mutants of *evgS* and *evgA* failed to confer GaNt tolerance. The regulation of GaNt tolerance by EvgS substitutional mutant (E701G) depends on phosphor-transfer from EvgS to EvgA. The phosphorylated EvgA up-regulates the transcription of *safA*, *ydeO*, and *gadE*. Deletion of *gadE* in the E701G mutant failed to confer GaNt tolerance, while deletion mutants of *ydeO* and *safA* showed partial reversal of tolerance. Thus, GadE acts as a key regulator of EvgS mediated GaNt tolerance and is the central regulator of glutamate-dependent acid resistance system [91].

#### 5.3.2. Activation of the KdpA Proton Pump

Potassium ions are needed for a variety of cellular functions, including intracellular pH regulation and cross-membrane potential. KdpA is a part of the KdpFABC ion channel involved in the ATP-driven transport of potassium ions across the cytoplasm [154]. Recent comparative studies on acid-adapted and non-adapted EHEC strains revealed activation of the KdpA proton pump in response to low pH [16]. During acid stress, this system blocks the flow of protons across the cell to increase the survival rate by more than 100 hrs [16]. Transcriptomic studies revealed upregulation of KdpA, BhsA (outer membrane protein), and ArnA in acid-adapted *E. coli* O157:H7 strain [16,155]. The enhanced expression of ArnA confers resistance to polymyxin B and colistin in growth cultures (Table 3). BhsA renders the outer membrane hydrophobic by modifying the lipopolysaccharides in a way that renders the cell surface more hydrophobic than hydrophilic [155]. This outer-membrane modification helps by-pass disruptive damage from cationic antimicrobial peptides and also increases cell aggregation [155]. These findings confirm that the KdpFABC ion channel regulates the development of antibiotic resistance in acid-adapted EHEC strains.

**Table 3 antibiotics-10-00522-t003:** Role of different two-component systems involved in mediating antibiotic resistance in response to acid stress. All abbreviations are listed at the end of the manuscript.

Treatment under Acid Stress	Two-Component Systems Involved	Acquired Antibiotic Resistance/Tolerance	Phenotypic Expression ^1^	Reference
Δ*tatC*, over-expressed *nlpE*	CpxRA	Cationic antimicrobial peptides (CAPs)	Increased tolerance	[145]
Δ*rcsF*, Δ*rcsB*, Δ*cpxR*	RcsCB, CpxRA	Mecillinam and cefsulodin	Increased tolerance	[139]
Δ*cpxR*	CpxRA	Cephalexin	Increased tolerance	[156]
W3110 *tol*C732::kan, W3110 *acr*B747::kan, W3110 *mdtB*774::kan, W3110 *mdtF*769::kan, W3110 *emrY*776::kan, W3110 *emrB*767::kan, W3110 *marR*751::kan	MarRAB, AcrAB, EmrKY, MdtABC and TolC	Extended-spectrum β-lactamases (ESBLs)	Increased tolerance	[152]
Δ*mar*	MarRAB, AcrAB and TolC	Beta-lactamase, rifampicin, spectinomycin, streptomycin, tetracycline, nalidixic acid	Increased resistance	[157]
Δ*rcsF* and Δ*rcsB*	RcsBC	Cefsulodin	Increased tolerance	[139]
	RcsBC, CpxRA, BaeSR	Mecillinam and cefsulodin		
*baeR* cloned pTrc99A plasmid	BaeRS, MdtABC, ArcAB	Ceftriaxone,	8 fold increased resistance	[158,159]
		novobiocin,		
		deoxycholate		
pH stress only	ArcAB, MarRAB	Ceftriaxone,	Presence ofhyper-resistantcolonies	[60]
		amikacin,		
		nalidixic acid		
	ArcAB, MarRAB, MdtABC	Multidrugresistance		[144,160]
	RcsCB	Cationic antimicrobial peptides (CAPs)	Intrinsic resistance	[161]
		Aztreonam		[162]
		Beta-lactams		[161]
		Daptomycin		[96,163,164]
Δ*dpiA*, Δ*cpxR*	RcsBC, CpxRA	Ampicillin	Increased tolerance	[165]
Δ*pmrA,* Δ*pmrB*	PmrAB, *arn* operon	PolymyxinB	Increased tolerance	[137]
Δ*acrB*	BaeSR, RcsBC, CpxRA, EvgAS, ArcAB	Multidrug resistance	16- to 32-fold increased resistance	[166]
Δ*marR*	MarRAB	Norfloxacin	Increased tolerance	[44].
Δ*acrB*Δ*evgAS*, Δ*acrB*Δ*emrKY*, Δ*acrB*Δ*yhiUV*Δ*emrKY*, Δ*acrB*Δy*hiUV*Δ*emrKY*/pUC*evgA*	ArcAB, EvgAS, EmrKY	Multidrug resistance	4 fold increased resistance	[167,168]
Overexpression of *baeR*, *evgA*, *rcsB*	BaeSR, RcsBC, CpxRA, EvgAS, ArcAB	Multidrug resistance	16- to 32-fold increased resistance	[166]

^1^ Resistance is an increase in MIC above the breakpoint; tolerance is the loss of killing with no change in MIC.

#### 5.3.3. Activation of TolC-Dependent Efflux Pumps

Nitrosative stress is a type of acid stress that is induced by the high concentration of nitric acid in the gastric fluid. This stress affects the transcription of several regulatory proteins [169]. EHECs activate several multidrug-resistance efflux pumps that contribute to both intrinsic and acquired antibiotic resistance [169,170].

##### AcrAB-TolC Regulation under Anaerobic Conditions

AcrAB-TolC is a resistance nodulation division (RND-type) efflux pump that contains an outer membrane channel (TolC), an inner membrane channel (AcrB), and a periplasmic protein (AcrA). As a housekeeping TCS, it is constitutively expressed to provide intrinsic resistance towards various toxins [130]. Under anaerobic conditions, AcrA (response regulator) triggers the upregulation of acid-induced efflux genes (*gadE-mdtEF* operon) by more than 20-fold (the activation of the *gadE-mdtEF* operon under aerobic conditions is controlled by EvgSA [130]). TolC is also reported to enhance GAD-EvgA acid tolerance, while bile salts and fatty acids present in the stomach trigger AcrAB-mediated activation of another global regulator *rob* [152,170]. This enhancing regulation of efflux genes results in increased drug resistance and survival of EHECs under nitrosative anaerobic environmental conditions in the human gut. Significantly reduced survival rate has been reported in knock-out mutant strains of MdtEF and MdtABC (BaeSR regulated efflux pump) [130,152]. Further studies confirmed that AcrAB-TolC deletion mutants showed attenuated colonization in mice and chickens [170]. In contrast, under extremely acidic conditions, knockout mutants of *tolC*, *emrB*, *mdtC*, and *mdtB* showed extremely low survival rates (Table 3) [152]. These findings suggest a role for efflux pumps in the development of multidrug resistance and enhanced survival rate for EHECs while passing through the stomach.

##### Activation of Multiple Antibiotic Resistance Operon

Members of the enterobacteriaceae family have a locus called the multiple antibiotic resistance (*marRAB* operon), which can confer cross-resistance to several antibiotics including tetracycline, ampicillin, norfloxacin, chloramphenicol, nalidixic acid, and β-lactams [171]. The MarR transcriptional regulator belongs to the AraC/XylS regulatory family that is responsible for inducing adaptive changes in response to environmental stress. Antibiotic resistance induced by the Mar operon is influenced by low pH-mediated acidification of the cytoplasm [44]. Experimental studies show that norfloxacin-sensitive, wild-type EHECs display a significantly enhanced norfloxacin-resistant phenotype when subjected to low pH (Table 4) [44]. Acid-triggered *mar* regulation also upregulates the transcription of *inaA*, whereas deletion mutants of this gene showed increased chloramphenicol and nalidixic acid resistance. Several studies also show that *inaA* is located within the *mar* locus [73,172,173,174,175,176].

##### BaeSR: Multidrug-Resistance Efflux Pump Regulator

BaeSR is one of the stress-triggered systems involved in the regulation of TolC-dependent multidrug efflux pumps (MdtABCD) and Spy (periplasmic chaperone) (Figure 3) [177,178,179]. As mentioned earlier, knockout mutants of *mdtABC* showed a significantly reduced survival rate under extremely acidic conditions (Table 3) [152]. As expected, Overexpression of the response regulator BaeR results in enhanced expression of MdtA and the AcrD efflux pump that mediates beta-lactam, cephalosporin and novobiocin resistance in large mammals (calves, pigs, and chickens) (Table 3 and Table 4) [151,179].

#### 5.3.4. Prophage-Encoded AraC-Like Transcriptional Regulators

EHEC serotype O157:H7 strains possess a locus of enterocyte effacement pathogenicity island (LEE-PAI), which regulates virulence genes of T3SS, intimin and Tir (translocation receptor) [59,61]. These genes are required to colonize, adhere to, produce intestinal lesions, and destroy intestinal microvilli. When EHECs occupy favorable environmental niches in the host intestine, LEE-PAI causes many virulence factors to be expressed [59]. Expression of LEE-PAI genes is tightly regulated by a set of transcriptional regulators (GadE, QseA, H-NS, IHF (integration host factor), Ler, and GrlA). Further studies showed that prophage-encoded loci of EHEC O157:H7 strains (specifically the EDL933 strain) carry a set of AraC-like transcriptional regulators PatE, PsrA, and PsrB. Mutational studies of these genes suggest that PatE and PsrB act as positive regulators of glutamate-dependent acid-resistance genes and trigger several key virulence determinants in acidic environments [59].

## 6. Acquired Antibiotic Resistance among EHEC Serotypes

In addition to acid tolerance, the glutamate-dependent acid resistance pathway performs several extended cross-protective functions, such as strengthening the cell envelope, enhanced attachment, colonization, biofilm formation, multidrug resistance, and bacterial pathogenicity [180,181]. Recent studies on multidrug-resistant EHECs confirm that under low pH stress, different serotypes respond variably toward acid resistance. This variation is attributed to the expression of glutamate-dependent regulation of RpoS [182]. RpoS deletion mutants of EDL933 and other O157:H7 strains show down-regulation of GadA and acid fitness island genes [183,184]. In comparison to other serotypes, the EHEC O157:H7 and EHEC O26:H11 strains show enhanced resistance and improved survival in the mammalian gut [185]. Transcriptomic profiles of virulent EHEC serotypes confirm that the expression level of *gadA*, *gadB,* and *gadE* genes is significantly upregulated when exposed to low pH [185]. As expected, an O157:H7 knockout mutant of the central regulator GadE resulted in 40-fold decreased expression of GadA and enhanced susceptibility towards acid stress. Likewise, knockout mutagenesis of EHEC strains confirmed a non-colonizing phenotype for *rcsB*, *arcA*, *cpxR*, excluding *evgS* in mouse models [126,127]. Surprisingly, ArcA induced expression of GadE-MdtEF is highly dependent on the anaerobic environment provided by the human stomach [131]. These findings confirm that EHECs cannot survive in the human gut in the absence of the acid-adapted regulatory changes [65,98,186,187].

Interestingly, these adaptations not only help the pathogens survive but also facilitate high virulence and enhanced colonization in animal models. Although mutant studies provide information about the adaptation, colonization, and infection pattern of EHECs, appropriate animal models still need to be developed [188]. For example, mouse models have a gastric pH is less acidic than that of humans [189,190]. Nevertheless, several statements can be made from other systems. For example, GadC deletion mutants of serotype O157:H7, when grown in a calf model, show reduced survival [191]. Moreover, low pH-pretreated *gadE* and *dctR* transposon mutants of O157:H7 show strong adherence to human epithelial type 2 and human colorectal adenocarcinoma cell lines, increasing apoptosis [15]. Other studies confirm an acid-induced YadK adhesin that allows EHEC to adhere strongly to epithelial cells and facilitate bacterial-host attachment, resulting in increased colonization and pathogenesis [192]. Multiple studies with EHEC O157:H7 and other serotypes have also reported the importance of acid-induced development of other phenotypes that enhance survival and improve virulence [44,87,90,91,94,129,132,185,188,193,194]. EHECs use low pH stress to adapt and regulate a wide array of genes that enhance survival and increase pathogenicity.

**Table 4 antibiotics-10-00522-t004:** Effect of low pH-mediated cross-protection against antibiotics and minimum inhibitory concentrations (MIC) of different EHEC serotypes.

Acid-Adapted Strains	pH	Acquired Resistance	MIC	Reference
EHEC O157:H7 ATCC 43889	2.75	Polymixin B, Colistin	Increased	[16]
*E. coli* ATCC25922	Acidic	Colistin	Increased	[195]
*E. coli* (EHEC) ATCC 43889*E. coli* ATCC 10536	2	Tetracycline	Increased	[41]
Foodborne EHEC strain	4	Nalidixic acid, amikacin, ceftriaxone	5 fold increase	[60]
*E. coli* K-12	2	Multidrug resistance	Increased	[45]
*E. coli* O157:H7 strain	4.8	Trimethoprim, ampicillin, and ofloxacin	Increased	[196]
EHEC Gut flora	2.5–4	Multidrug resistance	Increased	[43,197,198,199,200,201,202]
		Tetracycline		[48]
Rifampicin resistant *E. coli*	2.5–4	Sulphonamide, gentamicin and ampicillin	Increased	[203]
*E. coli* O157:H7	3.7	Streptomycin	Increased	[204]
29A and 29B EHEC strains	2.5–4	Ampicillin	Increased	[205]
*E. coli* IID 5208	3.2	Chitosan	Increased	[206]
Foodborne *E. coli*	Acidic	Aminoglycosides, cephalosporins, and quinolones	Increased	[207]
*E. coli* ATCC 12806	Acidic	Ampicillin-sulbactam, amoxicillin-clavulanic acid, cefotaxime, trimethoprim-sulphamethoxazole, tetracycline, ciprofloxacin, nitrofurantoin	Not evaluated	[208]
*E. coli* O157:H7	Acidic	Amoxicillin, tetracycline, ciprofloxacin, chloramphenicol, streptomycin, erythromycin, and gentamicin	Increased	[209]
*E. coli* BW25113	3	Trimethoprim	Increased	[46]
*E. coli* O157:H7*, E. coli* O26:H7	4.2–4.4	Ampicillin, kanamycin, streptomycin, trimethoprim, nalidixic acid, rifampicin, sulphonamides, chloramphenicol, chloramphenicol, tetracycline, minocycline, doxycycline	Increased	[210]
*E. coli* O157:H7	1.5	Trimethoprim, ampicillin, ofloxacin	Increased	[211]
*E. coli*	Acidic	Ampicillin	Increased	[212]
*EHEC* W3110	Acidic	Chloramphenicol	Increased	[47]
EHEC EV18 strain	Acidic	Norfloxacin	Increased	[44].
*E. coli* K12	Acidic	Cephalosporins, ceftiofur, cefotaxime	2-fold increased	[151]

Note: MIC above the breakpoint indicates that the organism is resistant.

## 7. Effect on Pathogenicity and Biofilm Formation

The molecular mechanisms underlying the pathogenic regulation of infectious *E. coli* indicate that biofilm formation correlates significantly with pathogenicity. Approximately 42 genes are regulated within a biofilm matrix in response to acid stress [213,214,215,216], including differential expression of *rpoS* [217] *gadAB*, *gadC*, *hdeABD* and *yjiD* (anti-adapter protein *iraD*, which inhibits *rpoS*). Knockout mutants of the genes mentioned above, when grown in glutamate-rich medium, increased biofilm formation [218]. Transcriptomic analysis of another gene cluster, *ymgABC,* revealed a significant role in regulating acid stress, with the *ymgB* gene product being downregulated in biofilm-forming cells [219]. To confirm a role in acid regulation, ten isogenic mutants of *E. coli* strain K-12 (Δ*ymgB*, Δ*ymgA*, Δ*ymgC*, Δ*ycgZ*, and Δ*gadB*, Δ*gadA*, Δ*gadE*, Δ*hdeB*, Δ*hdeA,* and Δ*hdeD*) were grown in glutamate enriched medium resulting in enhanced biofilm formation. These results highlight the importance of acid-resistance genes in biofilm formation [219].

Additionally, activation of several TCS response-regulators stimulates the expression of acid-fitness-island genes under acid stress that play an important role in pathogenesis regulation. In EHECs, NtrC, RcsB, and GadX are involved in the upregulation of the LEE (locus of enterocyte effacement) pathogenicity island, which indicates that nitrogen metabolism and glutamate-dependent-system genes play important roles in pathogenesis regulation [104,218,220]. Biofilm formation by another *E. coli* strain (MG1655) significantly increased at pH 5.5, while at lower pH the expression of flagellar synthesis genes and several virulence factors was strongly induced [1,221]. These studies highlight the biological relevance of acid stress in the regulation of pathogenesis in pathogenic *E. coli* [1,222,223].

## 8. Risk of Acquired Resistance in Non-Pathogenic Bacteria

Under respiratory stress, expression of the GAD operon is equally essential for pathogenic and non-pathogenic *E. coli* [82,224,225]. In an acidic environment, GadC consumes protons to promote GABA production that generates a proton motive force along with ATP production. Specifically, commensal bacteria and lactic acid bacteria (LAB) harbor GAD to produce GABA and act as probiotics in the GIT [224,226]. GABA plays an important role in bacteria that helps in the fermentation of protein-rich foods, such as cheese, rice germ, kimchi, yogurt, green tea, and sourdough [82,225,227]. Recent studies have found that fermentation of grapes by *Lactobacillus plantarum* DSM 19463 results in the production of GAD-derived GABA, which plays an important role in inducing the expression of β-defensin-2, hyaluronan synthase, and filaggrin genes responsible for skin protection in humans [228]. These remarkable findings lead to novel cosmetic formulations to treat antimicrobial problems related to skin.

Non-pathogenic bacteria maintain long-term commensalism with the host by stimulating the host immune system and inhibiting the colonization of gut pathogens [229,230,231]. To survive pH fluctuations in different compartments of the gut, the commensal bacteria also undergo the same extent of outer-membrane lipopolysaccharide modifications that contribute to ampicillin resistance. These changes result in modification of lipid A by LpxF phosphatase in commensal isolates of *Bacteroidetes thetaiotaomicron* that show significantly high polymyxin B resistance and enhanced colonization [232]. These adaptations in gut microbiota occur in response to environmental change. Clinical studies also report acquired tetracycline resistance in 22–33% EHECs in the gastric fluid by horizontal gene transfer, indicating an alarming health concern [41]. In some cases, commensal bacteria are reported to cause diseases, such as Crohn’s disease (CD), inflammatory bowel disease (IBD), and ulcerative colitis (UC) [233,234,235,236]. These adaptive pathogenic changes in gut microbiota occur in response to environmental factors, biodiversity, and genetic adaptability [230]. These findings indicate that modification of the lipid A anchor and other processes can promote a long-term commensal relationship between host and bacteria. On the other hand, horizontal gene transfer provides an open passage for the evolution of opportunistic pathogens having reduced antimicrobial susceptibility.

## 9. Conclusions and Future Perspective

EHECs have adapted to survive pre- and post-ingestion acid stress, thereby contributing to enhanced pathogenesis. Low pH positively regulates several metabolic pathways, such as motility, biofilm, chemotaxis, periplasmic secretory systems, and multidrug resistance that collectively regulate virulence. We highlighted the role of several signal-transduction cascades that enhance acid tolerance that results in the acquisition of antibiotic resistance. At present, almost all reported drugs are ineffective at controlling the spread of EHECs. Globally, the increasing resistance towards various classes of antibiotics, specifically cationic antimicrobial peptides and extended-spectrum β-lactamases, has become an overwhelming problem, making EHEC infections untreatable. EHECs have established complex regulatory mechanisms involving structural modification and efflux activation that provide an alarming condition for the emergence of new multidrug-resistant pathogens having improved colonization and infection capabilities.

Several factors influence the organism’s choice of the resistance mechanism. Cytoplasmic acidification offers a baseline level of defense that can act in tandem with modifications/mutations to reduce antibiotic susceptibility. Bacterial exposure to low pH is associated with acquired antimicrobial resistance to various therapeutic antibiotics. Active efflux and structural modifications of the bacterial membrane are the best-documented mechanisms responsible for bacterial cross-protection to antibiotics. The judicious and rational use of acidic treatments is crucial to reduce the risk of selecting antimicrobial-resistant bacteria. Antibiotic resistance acquisition strategies are extremely diverse; knowledge of this phenomenon at the molecular level provides an understanding of the details and appreciation to scale this important health problem. An even deeper understanding of the defensive responses deployed by the pathogens may reveal novel targets for agents that will help overcome the spread of foodborne diseases.

We emphasize that rigorous hygiene measures must be followed and all available antimicrobial agents should be used wisely to control the spread of multidrug-resistant strains. The risk of acid-adapted cross-protection by subsequent antimicrobial inactivation necessitates the identification of novel determinants that can influence the future epidemiology and health impact of multidrug-resistant infections. More efforts should be placed to develop novel non-antibiotic approaches such as vaccines, immuno-stimulants, phage therapies, prebiotics, and probiotics to treat EHEC infections.

## Figures and Tables

**Figure 1 antibiotics-10-00522-f001:**
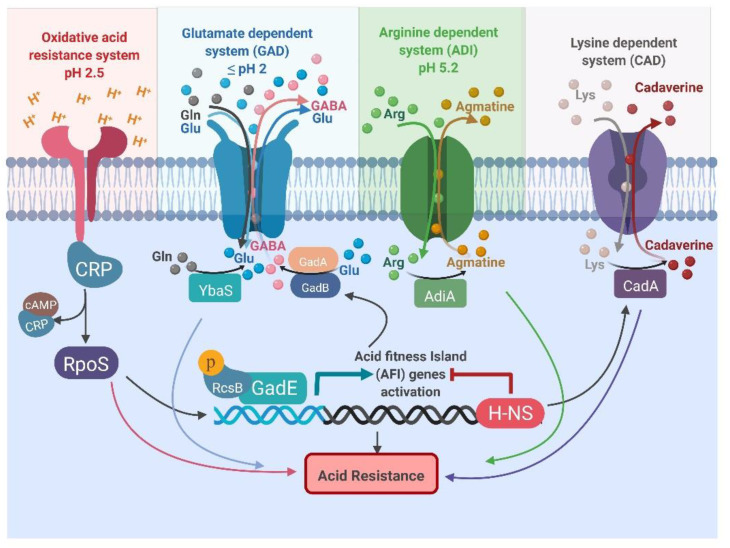
Representation of the oxidative, glutamate-dependent, arginine-dependent, and lysine-dependent acid resistance systems in *Escherichia coli*. All abbreviations are listed at the end of the manuscript.

**Figure 2 antibiotics-10-00522-f002:**
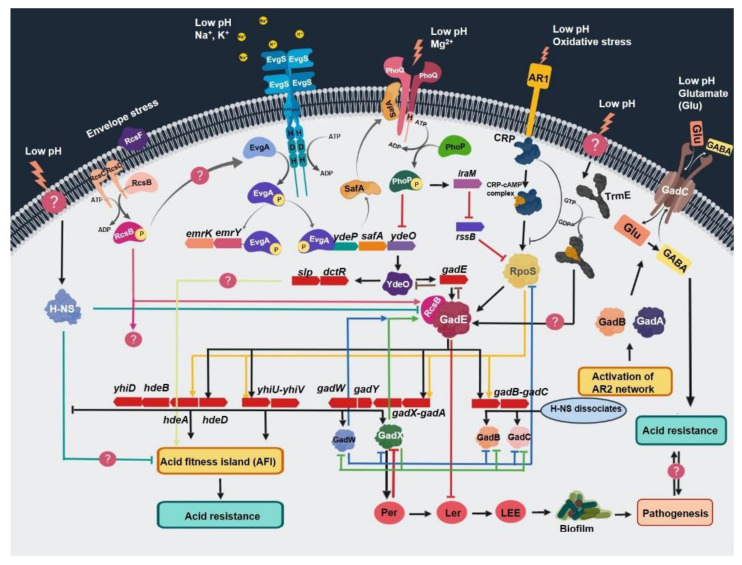
Schematic representation of acid stress regulation by different two-component signal transduction systems, acid-resistance networks, and their interconnecting assemblies. All abbreviations are defined at the end of the manuscript.

**Figure 3 antibiotics-10-00522-f003:**
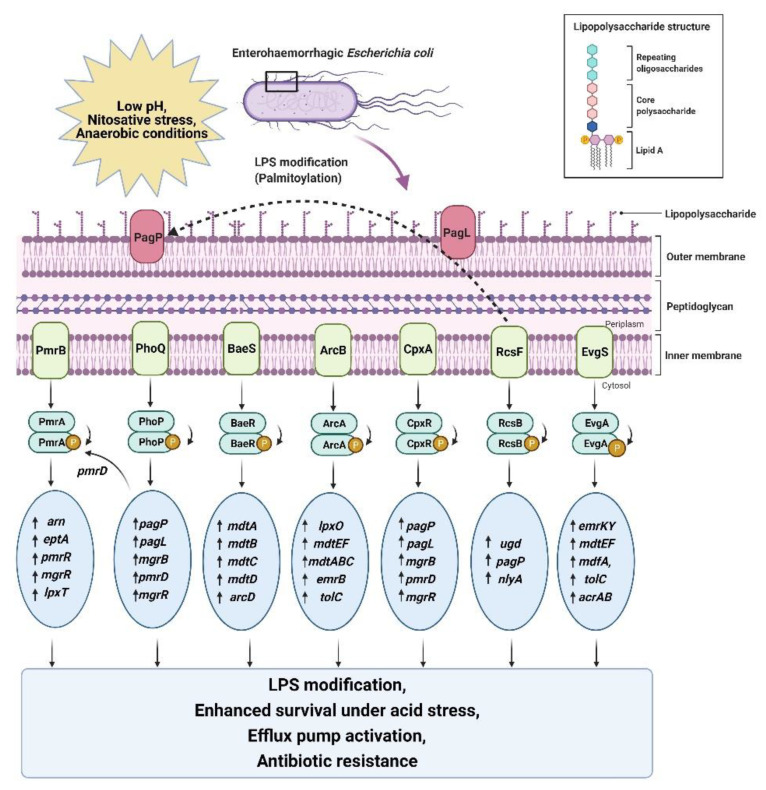
Schematic representation of the acid-induced activation of a specific set of genes by two-component systems leading to LPS modification, efflux pump activation, enhanced survival, and antibiotic resistance in EHECs. All abbreviations are defined at the end of the manuscript.

**Table 2 antibiotics-10-00522-t002:** Acid-adapted pathogens other than EHEC pathotypes’ and their acquired antibiotic resistance.

Organism	Treatment	Acquired Resistance	MIC at Low pH	Reference
Listeria monocytogenes	pH 5.5–6.0	Multidrug-resistant	Increased	[119]
	Acid stress	Erythromycin, ciprofloxacin, nitrofurantoin	Increased	[120]
Salmonella enterica	pH 2–3.8	Tetracycline, chloramphenicol, ampicillin, penicillin, cephalosporins, ceftriaxone, cefepime, kanamycin, gentamicin; ciprofloxacin, cyclic lipopeptide polymyxin, sulfamethoxazole-trimethoprim	Increased	[121]
		Chloramphenicol, tetracycline, ampicillin, acriflavine, triclosan	Increased	[122]
Acinetobacter baumannii	Acid stress	Amikacin, norfloxacin, imipenem, meropenem piperacillin-tazobactam	Increased	[123]
Cronobacter sakazakii	pH 3.5	Tetracycline, tilmicosin, florfenicol, amoxicillin, ampicillin, vancomycin, neomycin, ciprofloxacin, enrofloxacin	Increased	[124]
Staphylococcus aureus	pH 1.5	Multidrug-resistant	Increased	[125]

Note: MIC above the breakpoint indicates that the organism is resistant.

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
