# Peer review of "Insights into Emergence of Antibiotic Resistance in Acid-Adapted Enterohaemorrhagic Escherichia coli"

_antibiotics, 2021, doi:10.3390/antibiotics10050522_

Round 1

Reviewer 1 Report

  • This review addresses different metabolic adaptations to acidic conditions which could enhance the acquisition of antibiotic resistance in EHEC. The article gives an interesting scientific perspective on a field that represents a serious concern all over the world. This is a review with a significant impact on the scientific community interested in antimicrobial-resistance mechanisms and fits within the scope of the "Antibiotics" Journal. Therefore, it could be adequate for publication in this Journal. However, authors must carry out some improvements on the background and the text clearness/correctness. It is strongly suggested that the manuscript is checked by a native English speaker.
  • In the following lines, this reviewer gives some suggestions for the authors' consideration. Moreover, some other points should be addressed:

-Title: the manuscript is particularly dealing with EHEC by which "foodborne Escherichia coli" must be replaced by "Enterohaemorrhagic Escherichia coli"

-Lines 63-64, "EDL933" is not a serotype but a EHEC strain. A reference is required.

-Lines 28-29, consider rephrasing.

-Line 94, introduce properly "EDL933" (EHEC strain) and clearly state its importance since it is cited several times in the review. Write it correctly here and beyond.

-Line 93, "Pathogenic EHECs" is redundant. According to this group definition, all their members are pathogenic bacteria. Also revise the remaining cites all over the text.

-Lines 93-95, the sentence must be rephrased.

-Figure 1: A list of abbreviations is required (close to the Figure or any place in the article; additionally, it should include abbreviations for the other tables/figures). Moreover, a better explanation of that showed in the figure must be provided in the text.

-Consider homogenising figure and table captions (Is the first letter of each word in capital? Please, unify).

-Figure 2 caption: it seems to be too long. Consider splitting and add extra information (explanation) in the text.

-Lines 208-240. Why are these regulator systems revised? A justification should be included in the chapter.

-Revise all subsection numbers (e.g. 2.3.1..., 4.3.3....). Currently, it is a mess.

-Table 2, improve the format of the table by making differences between rows according to the columns.

-Line 311, delete "Gram-negative".

-Line 326, correct "envelop".

-Figure 3 caption, in a model bacteria? Consider including more details.

-Line 414, delete "foodborne".

-Line 453, "Few studies... [50]". Please, include the studies.

-Lines 476 and 540, "...Acid-Adaptive Antibiotic Regulation..". Consider rephrasing.

-Line 574-575: it seems not to be appropriated for a conclusion section.

-Conclusions and future perspective, it is not well-stated gaps in the field which provide new challenges for researchers shortly. So, consider adding explicitly future perspectives.

Finally, correct or reword the following (not being exhaustive): line 67 ("to date"), lines 82-83 ("consuming contaminated food..."), line 92 ("posing a serious threat"), line 117 (two components i.e. ...), line 250 ("several other studies"), line 251 ("pyla's").

Author Response

The authors are very thankful to you for giving your valuable time to review the manuscript and for your kind suggestions to improve the manuscript.

We have revised the manuscript as per your comments and have made changes according to your suggestions. It is mentioned that due to the rephrasing and modifications made in the revised version of the manuscript as per reviewers’ instructions, the line numbers have been changed as compared to previous version. Please refer to new line numbers where we have made corrections respective to the lines mentioned by you in your comments. We have mentioned the new line numbers in each response for your convenience. Following are the point to point responses to your comments:

Point 1: Title: the manuscript is particularly dealing with EHEC by which "foodborne Escherichia coli" must be replaced by "Enterohaemorrhagic Escherichia coli" Response 1: The title of the manuscript has been modified as per your suggestion (line 3). Point 2: Lines 63-64, "EDL933" is not a serotype but a EHEC strain. A reference is required Response 2: The correction has been made in line 70 as per your instructions and the new references have been added to make it clear for the reader (line 72). Point 3: Lines 28-29, consider rephrasing. Response 3: We have rephrased the sentence as per your suggestion (lines 29-31). Point 4: Line 94, introduce properly "EDL933" (EHEC strain) and clearly state its importance since it is cited several times in the review. Write it correctly here and beyond. Response 4: The correction has been made (lines 119-120). We have explained EDL933 properly and clearly in lines 75-78 and the correction has also been done in the text (lines 70, 120, 146, 535 and 548). Point 5: Line 93, "Pathogenic EHECs" is redundant. According to this group definition, all their members are pathogenic bacteria. Also revise the remaining cites all over the text. Response 5: The correction has been made as per your instructions (lines 96, 119, 301, 559, 647 and 650). Point 6: Lines 93-95, the sentence must be rephrased. Response 6: The sentences have been rephrased as per your suggestion (lines 119-123). Point 7: Figure 1: A list of abbreviations is required (close to the Figure or any place in the article; additionally, it should include abbreviations for the other tables/figures). Moreover, a better explanation of that showed in the figure must be provided in the text. Response 7: All abbreviations have been added in the list of abbreviations at the end of the manuscript (line 691). This list includes all abbreviations used in tables, figures and text. The explanation of figures in text has also been updated as per your suggestion (lines 175-186 and 237-246). Point 8: Consider homogenising figure and table captions (Is the first letter of each word in capital? Please, unify). Response 8: As per your suggestion, captions of all figures and tables are homogenized. Point 9: Figure 2 caption: it seems to be too long. Consider splitting and add extra information (explanation) in the text. Response 9: The extra text has been deleted and relevant literature has been added in the text (lines 237-246). Point 10: Lines 208-240. Why are these regulator systems revised? A justification should be included in the chapter. Response 10: We have reviewed the role of evgAS, phoPQ and rcsB because the EHECs share these genes (same set of genes) to develop antibiotic resistance in response to low pH stress. Therefore it is important to discuss these systems to understand their mechanisms in detail. Supporting text has been added (lines 148-157) Point 11: Revise all subsection numbers (e.g. 2.3.1..., 4.3.3....). Currently, it is a mess. Response 11: As per your suggestion all subsection numbers are revised. Point 12: Table 2, improve the format of the table by making differences between rows according to the columns. Response 12: The table has been formatted to make it clear for the readers. Point 13: Line 311, delete "Gram-negative. Response 13: As per your suggestion, the word “Gram-negative” has been deleted (line 369). Point 14: Line 326, correct "envelop. Response 14: As per your suggestion, the correction has been done (line 384). Point 15: Figure 3 caption, in a model bacteria? Consider including more details. Response 15: We have modified the caption of the figure as per your suggestion. Point 16: Line 414, delete "foodborne". Response 16: We have made correction as per your suggestion (line 475). Point 17: Line 453, "Few studies... [50]". Please, include the studies. Response 17: We have added new references to support the text (line 518). Point 18: Lines 476 and 540, "...Acid-Adaptive Antibiotic Regulation..". Consider rephrasing. Response 18: As per your suggestions both headings have been rephrased (line 540 and 605). Point 19: Line 574-575: it seems not to be appropriated for a conclusion section. Response 19: We have deleted the sentence from the conclusion section (line 640-641). Point 20: Conclusions and future perspective, it is not well-stated gaps in the field which provide new challenges for researchers shortly. So, consider adding explicitly future perspectives Response 20: The modifications have been done as per your suggestion (lines 655-679). Point 21: Finally, correct or reword the following (not being exhaustive): line 67 ("to date"), lines 82-83 ("consuming contaminated food..."), line 92 ("posing a serious threat"), line 117 (two components i.e. ...), line 250 ("several other studies"), line 251 ("pyla's"). Response 21: The corrections have been done (lines 74-75, 103-104, 117-118, 158 and 304-306).

Reviewer 2 Report

In this paper entitled “Insights into Emergence of Antibiotic Resistance in Acid-Adapted Foodborne Escherichia coli”, Dr. Fei Shang, Dr. Ting Xue and co-workers review the emergence of multidrug-resistant pathogens, focusing on the water and foodborne Entero-haemorrhagic Escherichia coli.

They discuss the correlation between acid tolerance and antibiotic resistance; summarize the molecular mechanisms used by acid-adapted EHECs, and highlight low pH stress as an emerging player to develop cross-protection against antimicrobial peptides. Finally, they underline the need to explore potential gene targets to overcome the risk of multidrug-resistant foodborne diseases in healthcare and industry.

The paper is interesting, clearly written, the data are nicely presented and support their conclusions.

Author Response

The authors are very thankful to you for giving your valuable time to review the manuscript and for your kind suggestions to improve the manuscript. It’s a pleasure to know that you find our manuscript entitled, “Insights into Emergence of Antibiotic Resistance in Acid-Adapted Enterohaemorrhagic Escherichia coli” interesting and nicely presented.

Reviewer 3 Report

This review describes the current literature on the relationships between acid adaptation mechanisms and antibiotic resistance in EHEC, the molecular mechanisms involved, and its relationship with the prevention and treatment of food borne diseases. Overall, the review is well written (although its appeal to food microbiologists may be limited), but there are a few things that should be clarified.

l18 what do you mean? this sentence is ambiguous... Low pH AND organic acids (or other acid preservatives) can slow growth and/or inactivate vegetative cells, depending on dose and exposure time. Please rephrase

l27 only peptides?

l35-36 this is a (generic) range of growth; maximum or near maximum growth rates are associated with a much narrower pH range; please rephrase

l38-39 are you sure? this is highly irritating and, while this is not my field, I think other disinfectants are used for wounds

l41 I doubt that "sprinkle" is the right term for produce; electrolysed water and chlorine solutions are used for the disinfection of wash water

l43 facilitate?

l45 do you mean "spread of antibiotic resistant EHEC strains"? There are other reasons for the spread of EHEC in animal husbandry

l46 unclear, please rephrase; are you making reference to acidosis? This is not related to administration of acids...

l49 I am not sure the surface of spinach leaves can be defined as an acidic environment; please check

l66-68 can you expand a little bit; I take you are referring to a particular clonal complex here

l74 and transposons? Bacteriophages are also major vehicles for the transfer of pathogenicity islands

l79 processing and cooking

l85 I think eradicate is a bit strong; in fact, lethality would depend on pH and residence time

l88-91 unclear, please reformulate and may be explained a little bit. Making reference to papers/review is not enough for such a strong concept...

l94 and elsewhere, I doubt that "adaptive strains" is good English usage (however, English is not my first language)

l97-98 this is too generic to be true; I can think of many antimicrobial agents which have other mechanisms of action; please expand

l99-100 that's only one component (there are also proteins and phospholipids...)

Fig 1 these are antiporters; I would redraw adding a (curved) arrow showing the export of the amines in exchange of the amino acids

l243-346 very unclear. Are you listing mechanisms for cross-protection? Please rephrase.

Table 2 is difficult to read; I would add horizontal borders to separate the effect for different microorganisms.

l261-262 unclear; what do you mean by "consume protons"; the outer membrane and the cytoplasmic membrane in absence of ionophores are impermeable to protons; protons are co-transported with organic acids and which then dissociate in the cytoplasm...

l272 and thrive during acid stress? May be thrive is not the right verb here...

l274-277 very unclear, please rephrase

l284 what do you mean by stronger?

l356-57 this is a bit awkward, please rephrase

l386 what do you mean by "monitors"?

l417 I think the first sentence can be removed; it is repetitious

l496-497 "facilitates low infectious dose" please rephrase

l500-502 do you mean human? Mice and rats are, indeed, mammalians; what do you mean by risk of contaminating the food chain? In an experiment using cows/beef you do not need to use milk or meat for food, you can just destroy it;

table 4 same problem as table 2; please use horizontal borders between rows as in table 3

l540 the title of this paragraph is a bit awkward, pleas rephrase

l551 is this a beta?

l555 please avoid using microflora, microbial flora etc.

l556-557 do you mean in the GIT? in a given section of the GIT pH tends to be rather constant, with the exception of the upper GIT

l565-567 unclear, please rephrase

l574 the notorious group?

Author Response

The authors are very thankful to you for giving your valuable time to review the manuscript and for your kind suggestions to improve the manuscript.

We have revised the manuscript as per your comments and have made changes according to your suggestions. It is mentioned that due to the rephrasing and modifications made in the revised version of the manuscript as per reviewers’ instructions, the line numbers have been changed as compared to previous version. Please refer to new line numbers where we have made corrections respective to the lines mentioned by you in your comments. We have mentioned the new line numbers in each response for your convenience. Following are the point to point responses to your comments:

Point 1: L18 what do you mean? this sentence is ambiguous... Low pH AND organic acids (or other acid preservatives) can slow growth and/or inactivate vegetative cells, depending on dose and exposure time. Please rephrase

Response 1: The modification has been done (line 18). To kill bacteria we can use acid-based disinfectants (which kill a wide range of microbes on non-living surfaces to prevent the spread of diseases), antiseptics (which are applied to living tissue and help reduce infection during surgery), and antibiotics (which destroy microorganisms within the body). While antibacterial agents can be further subdivided into bactericidal agents, which kill bacteria, and bacteriostatic agents, which slow down or stall bacterial growth.  

Point 2: L27 only peptides?

Response 2: The correction has been done. The word “peptide” has been replaced with “agents” (line 27).

Point 3: L35-36 this is a (generic) range of growth; maximum or near maximum growth rates are associated with a much narrower pH range; please rephrase

Response 3: The correction has been done and a supporting reference has been added (line 38).  

Point 4: L38-39 are you sure? this is highly irritating and, while this is not my field, I think other disinfectants are used for wounds.

Response 4: Supporting text has been added to clarify the statement and relevant references have also been added (lines 41-43).

 Point 5: L41 I doubt that "sprinkle" is the right term for produce; electrolysed water and chlorine solutions are used for the disinfection of wash water.

Response 5: We have replaced the word sprinkle with sprayed (lines 44-45).

Point 6: L43 facilitate?

Response 6: We have rephrased the lines 46-47 to make it more clear and understandable.

Point 7: L45 do you mean "spread of antibiotic resistant EHEC strains"? There are other reasons for the spread of EHEC in animal husbandry.

Response 7: The excessive use of inappropriate antibiotics/growth promotors in animal husbandry and farming is the foremost reason for the spread of EHEC strains (lines 49-51). There are other reasons as well but this is the primary reason for the spread of antibiotic resistance. Other reasons are explained in lines 51-54.

Point 8: L46 unclear, please rephrase; are you making reference to acidosis? This is not related to administration of acids...

Response 8: We have rephrased the sentence to make it clear to the readers (line 52).

Point 9: L49 I am not sure the surface of spinach leaves can be defined as an acidic environment; please check.

Response 9: The correction has been made as per your suggestion (line 55).

Point 10: L66-68 can you expand a little bit; I take you are referring to a particular clonal complex here.

Response 10: We have added new reference and also mentioned the specific serotype which is reported in literature (lines 72-75).

Point 11: L74 and transposons? Bacteriophages are also major vehicles for the transfer of pathogenicity islands

Response 11: Supportive literature has been added to make it clear to the readers (lines 87-95).

Point 12: L79 processing and cooking

Response 12: We have rephrased the sentence to make it clear to the readers (lines 99-100).

 Point 13: L85 I think eradicate is a bit strong; in fact, lethality would depend on pH and residence time.

Response 13: As per you suggestion, we have changed the word in line 107.

Point 14: L88-91 unclear, please reformulate and may be explained a little bit. Making reference to papers/review is not enough for such a strong concept....

Response 14: We have added new literature to make it understandable to the readers. New references have also been added (lines 112-118).

Point 15: L94 and elsewhere, I doubt that "adaptive strains" is good English usage (however, English is not my first language)

Response 15: We have searched for more appropriate word in the reported literature but didn’t find any other suitable word. In all papers they have reported “adapted strains” which develop adaptive strategies to cope up with environmental stresses. So we have modified the word “adaptive strains” to “adapted strains” at all places in the text to make is more understandable for the readers (lines 110, 120, 139, 305, 309, 465, 469, 475, 560 and 668).

Point 16: L97-98 this is too generic to be true; I can think of many antimicrobial agents which have other mechanisms of action; please expand.

Response 16: As per your suggestions, we have added the literature about different mechanisms opted by antimicrobial agents and highlighted the importance of outer membrane in protecting the organism. Relevant references have also been added (lines 124-129).

Point 17: L99-100 that's only one component (there are also proteins and phospholipids...)

Response 17: As per your suggestion, we have changed the text and added new information (lines 131-133).

Point 18: Fig 1 these are antiporters; I would redraw adding a (curved) arrow showing the export of the amines in exchange of the amino acids.

Response 18: The curved arrows have been drawn to show the export of the amines in exchange of the amino acids.

Point 19: L243-346 very unclear. Are you listing mechanisms for cross-protection? Please rephrase.

Response 19: We have rephrased this section as per your instruction to make it clear for the readers. Lines 295-308 explain the development of cross protection in EHECs in response to environmental stresses. Lines 310-334 explain the general metabolic changes reported when EHECs suffer acid stress. Lines 335-539 explain the strategies (3 ways: 5.1, 5.2 and 5.3) opted by EHECs to develop cross protection against antimicrobial agents.

 Point 20: Table 2 is difficult to read; I would add horizontal borders to separate the effect for different microorganisms.

Response 20: Horizontal borders have been drawn to make it clear for the readers.

Point 21: L261-262 unclear; what do you mean by "consume protons"; the outer membrane and the cytoplasmic membrane in absence of ionophores are impermeable to protons; protons are co-transported with organic acids and which then dissociate in the cytoplasm...

Response 21: We have rephrased the sentence to make it clear for the reader and new references have also been added (Lines: 317-318).

Point 22: L272 and thrive during acid stress? May be thrive is not the right verb here...

Response 22: As per your suggestion the word “thrive” has been deleted (line 328).

Point 23: L274-277 very unclear, please rephrase

Response 23: We have rephrased the sentence and removed the mistakes (lines 331-333).

Point 24: L284 what do you mean by stronger?

Response 24: We have rephrased the sentence to make it clear for the reader (lines 341-343).

Point 25: L356-57 this is a bit awkward, please rephrase.

Response 25: We have rephrased the above mentioned lines to make it clear for the reader (lines 416-420).

Point 26: L386 what do you mean by "monitors"?

Response 26: We have rephrased the sentence to remove the mistakes (line 448).

 Point 27: L417 I think the first sentence can be removed; it is repetitious

Response 27: As per your suggestion, we have deleted the first sentence (line 481-482).

Point 28: L496-497 "facilitates low infectious dose" please rephrase

Response 28: As per your suggestion, we have modified this sentence to make it clear for the reader (line 562).

Point 29: L500-502 do you mean human? Mice and rats are, indeed, mammalians; what do you mean by risk of contaminating the food chain? In an experiment using cows/beef you do not need to use milk or meat for food, you can just destroy it;

Response 29: As per your suggestion, we have modified these sentences (lines 565-567).

Point 30: table 4 same problem as table 2; please use horizontal borders between rows as in table 3

Response 30: Horizontal borders are drawn to make it clear for the readers.

Point 31: L540 the title of this paragraph is a bit awkward, pleas rephrase

Response 31: As per your suggestion, we have modified the title of this paragraph (line 605).

Point 32: L551 is this a beta?

Response 32: Yes it is beta, the beta sign has been added (line 617).

Point 33: L555 please avoid using microflora, microbial flora etc.

Response 33: As per your suggestion, we have modified this sentence (line 621).

Point 34: L556-557 do you mean in the GIT? in a given section of the GIT pH tends to be rather constant, with the exception of the upper GIT

Response 34: As per your suggestion, we have modified the sentence, as there is a fluctuation of pH in different GIT compartments while in each compartment it remains constant (line 623).  

Point 35: L565-567 unclear, please rephrase

Response 35: As per your suggestion, we have rephrased this sentence (line 631-632).

Point 36: L574 the notorious group?

Response 36: We have deleted the line as it was inappropriate for the conclusion section as suggested by the other reviewer (line 640-641)

Round 2

Reviewer 1 Report

The revised version of the manuscript has been considerably enriched.

Line 309 (table 2), it could be confusing to indicate "non-EHEC pathogens" if authors mean "pathogens other than E. coli pathotypes". Consider revising.

Author Response

We are very thankful to you for giving your valuable time to review the manuscript and for your kind suggestions to improve the manuscript.